# WHEN UNSUPERVISED DOMAIN ADAPTATION MEETS ONE-CLASS ANOMALY DETECTION: ADDRESSING THE TWO-FOLD UNSUPERVISED CURSE BY LEVERAGING ANOMALY SCARCITY

## ABSTRACT

This paper introduces the first fully unsupervised domain adaptation (UDA) framework for unsupervised anomaly detection (UAD). The performance of UAD techniques degrades significantly in the presence of a domain shift, difficult to avoid in a real-world setting. While UDA has contributed to solving this issue in binary and multi-class classification, such a strategy is ill-posed in one-class UAD. This might be explained by the unsupervised nature of the two tasks, namely, domain adaptation and anomaly detection. Herein, we first formulate this problem that we call the two-fold unsupervised curse. Then, we propose a pioneering solution to this curse, considered intractable so far, by assuming that anomalies are rare. Specifically, we leverage clustering techniques to identify a dominant cluster in the target feature space. Posed as the normal cluster, the latter is aligned with the source normal features. Specifically, given a one-class source set and an unlabeled target set composed primarily of normal data and some anomalies, we fit the source features within a hypersphere while jointly aligning them with the features of the dominant cluster in the target set. The paper provides extensive experiments and analysis on common domain adaptation benchmarks, adapted to the one-class anomaly detection setting, demonstrating the relevance of both the newly introduced paradigm and the proposed approach. The code will be made publicly available.

## 1 INTRODUCTION

Anomaly Detection (AD) can be seen as the identification of outliers deviating from a usual pattern. The growing interest in AD in both academia and industry is mainly due to its relevance in numerous practical scenarios, such as early disease detection in medical imaging (Huang et al., 2024; Bao et al., 2024) and industrial inspection (Sun et al., 2024; Mejri et al., 2024; Deng & Li, 2022; Beul et al., 2018; Liu et al., 2024). By definition, anomalies rarely occur. Annotating anomalous data is, therefore, often difficult and costly (Hermary et al., 2025; Sun et al., 2024; Cao et al., 2023), hindering the collection of large-scale datasets. As a result, state-of-the-art methods mostly tackle AD as an unsupervised problem (Han et al., 2022; Ruff et al., 2021), where the objective is to learn only from the normal class.

Despite achieving promising results, recent approaches in AD (Ruff et al., 2018; Deng & Li, 2022; Tien et al., 2023; Sträter et al., 2024; Hermary et al., 2025) typically assume that training and inference data are drawn from the same distribution. This assumption does not always hold in unconstrained scenarios, where a *domain shift* (Quinonero-Candela et al., 2022) between training and testing data can naturally arise due to varying setups, such as different lighting conditions and variations in object pose (Cao et al., 2023). As a result, a model trained on a dataset sampled from a given domain, usually called *source* dataset, will show degraded performance when tested on a dataset from a different domain, generally termed *target* dataset. For instance, an AD model for medical imaging trained on images acquired using a given Magnetic Resonance Imaging (MRI) device can fail to generalize to samples captured with a different MRI system.

To reduce such a domain gap while avoiding costly annotation efforts, Unsupervised Domain Adaptation (UDA) (Wilson & Cook, 2020; Kalluri et al., 2024) has proven to be an effective solution in binary and multi-class classification tasks (Singh et al., 2024; Kalluri et al., 2024). UDA aims at learning domain-invariant features by relying on labeled source and unlabeled target data at the same time. However, the task of **unsupervised** domain adaptation for **unsupervised** anomaly detection (UAD) is ill-posed as the goal is to: *align the source and the target feature distributions using only normal source data and unlabeled target data formed by both normal and anomalous samples* (see Figure 2 (c)). Hence, a direct extension of standard UDA

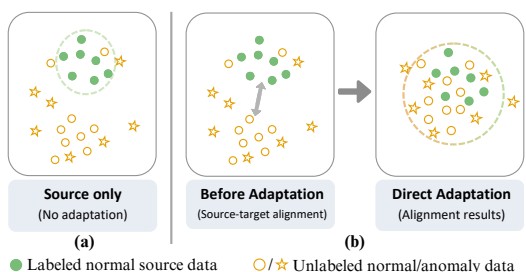

Figure 1. **Illustration of the two-fold unsupervised curse**: **(a)** The decision boundary learned from the source set without any adaptation does not allow generalization to the target domain. **(b)** Direct alignment of the unlabeled target with the one-class source features leads to the confusion of normal and abnormal samples.

techniques developed for binary/multi-class classification (Kalluri et al., 2024; Wilson & Cook, 2020) would not be applicable as these methods usually aim at minimizing the distance between the estimated distributions from the entire source and target training sets. Indeed, this would lead to the erroneous alignment of both normal and anomalous target samples with normal source samples, as illustrated in Figure 1 (b). Given the learned decision boundary, this would lead to the confusion of normal and abnormal samples from the target set. As it involves solving two unsupervised tasks simultaneously, we term this problem the *two-fold unsupervised curse*.

To the best of our knowledge, no prior work has tried to address this two-fold unsupervised challenge, i.e., unsupervised domain adaptation for one-class image anomaly detection described in Figure 2 (c). Indeed, related works have mainly simplified the problem by either (1) assuming the availability of labeled abnormal and normal source data, resulting in UDA for a binary classification setting (Kumagai et al., 2019) (see Figure 2 (a)), or (2) maintaining the source one-class setup while accessing only few normal target data referred to as few-shot supervised adaptation for unsupervised anomaly detection (Kumagai et al., 2019; Cohen & Wolf, 2019; Li et al., 2023; 2022; Yang et al., 2023) (see Figure 2 (b)). Nevertheless, annotating even a few samples might still be constraining, particularly in the field of anomaly detection, where expert knowledge is often needed, such as for tumor annotation in medical images (Huang et al., 2024; Bao et al., 2024) or for industrial inspection (Deng & Li, 2022; Tien et al., 2023; Liu et al., 2024). Moreover, few-shot adaptation approaches are known to be prone to overfitting issues since few shots cannot fully represent the normal target distribution (Song et al., 2023). This calls for a fully unsupervised domain adaptation approach that leverages the diversity of the available large, unlabeled target datasets.

In this paper, we investigate whether the rare occurrence of anomalies could be exploited to address the two-fold unsupervised curse. We herein propose the first unsupervised domain adaptation framework for unsupervised image anomaly detection. Our solution starts by identifying a dominant cluster assumed to be formed by normal target data and then aligning it with normal source samples. Specifically, our method utilizes a trainable ResNet-based (He et al., 2016) feature extractor to process both the source and target features. A frozen CLIP visual encoder (Radford et al., 2021) is also used to generate corresponding target features, which are then clustered using K-means to identify the samples of the dominant cluster. These samples are mapped into the ResNet-based (He et al., 2016) feature space and aligned with the source features. For the domain adaptation task, a contrastive strategy (Radford et al., 2021; Oord et al., 2018) ensures the similarity between the dominant target cluster and normal source samples, while for the anomaly detection task, a Deep Support Vector Data Description (DSVDD) (Ruff et al., 2018) objective enforces feature compactness on the normal source data. Our framework is modular, allowing for flexible component changes, and supports various adaptation strategies, including statistical and adversarial alignment. Experiments on standard UDA benchmarks (Saenko et al., 2010; Venkateswara et al., 2017; Peng et al., 2017; Li et al., 2017) for *semantic anomaly detection* (Sträter et al., 2024) demonstrate its effectiveness. Our method achieves state-of-the-art (SoA) performance, even against few-shot adaptation methods.

**Contributions.** The main contributions of this work can be summarized as follows: (1) The two-fold unsupervised curse of UDA for one-class anomaly detection is formalized, and the induced challenges are outlined. (2) A solution to the two-fold unsupervised problem is proposed by leveraging an

intrinsic property of anomalies, i.e., their scarcity. (3) A UDA method for one-class semantic anomaly detection is introduced, leveraging a Vision Language Model, namely CLIP (Radford et al., 2021), for dominant cluster identification and alignment using a contrastive strategy. (4) Extensive experiments and analysis are conducted on several benchmarks (Saenko et al., 2010; Venkateswara et al., 2017; Peng et al., 2017; Li et al., 2017), demonstrating the relevance of the proposed framework under both fully unsupervised and few-shot adaptation settings.

**Paper Organization.** Section 2 reviews UAD works under domain shift. Section 3 defines the two-fold unsupervised curse, while Section 4 and Section 5 detail one possible solution for solving it. Section 6 and Section 7 cover the experiments and limitations of this method. Section 8 concludes and outlines future work.

## 2    RELATED WORKS: ANOMALY DETECTION UNDER DOMAIN SHIFT

Unsupervised image anomaly detection is a well-established research area (Han et al., 2022; Ruff et al., 2021; Hermary et al., 2025; Ruff et al., 2018; Deng & Li, 2022; Tien et al., 2023; Sträter et al., 2024) where the aim is to learn a function $\zeta$ using a single class corresponding to normal data from the normal-only dataset $\mathcal{D}^n = \{(\mathbf{X}_i, y_i); y_i = 0\}_{i=1}^N$, to classify whether an input image $\mathbf{X}$ is normal ($y = 0$) or not ($y = 1$). This is achieved by optimizing the objective,

$$\min_{\zeta} \mathbb{E}_{(\mathbf{X}_i, y_i) \sim \mathcal{D}^n} \left[ \mathcal{L}\left( \zeta(\mathbf{X}_i), y_i = 0 \right) \right], \quad (1)$$

where $\mathcal{L}$ is a loss enforcing feature compactness as in DSVDD (Ruff et al., 2018) or a reconstruction loss typically used in autoencoders-based methods (Deng & Li, 2022; Tien et al., 2023). Although achieving impressive performance on standard benchmarks, the majority of AD methods (Han et al., 2022; Ruff et al., 2021; Hermary et al., 2025; Ruff et al., 2018; Deng & Li, 2022;

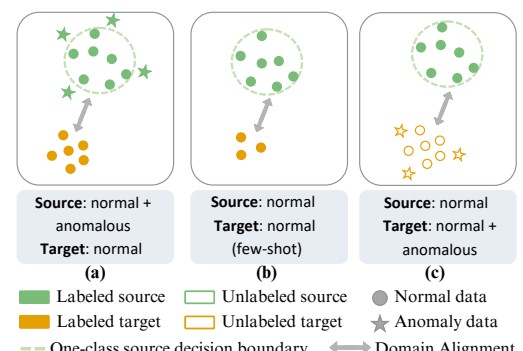

Labeled source / Unlabeled source / Normal data
Labeled target / Unlabeled target / Anomaly data
One-class source decision boundary / Domain Alignment

Figure 2. **Comparison of our domain adaptation with anomaly detection setting with previous works**: **(a)** supervised source anomaly detection with supervised domain adaptation (Kumagai et al., 2019), **(b)** unsupervised one-class source anomaly detection with few-shot domain adaptation (Li et al., 2023; Yang et al., 2023; Li et al., 2022), **(c) our setting**: unsupervised one-class source anomaly detection with unsupervised domain adaptation.

Tien et al., 2023; Sträter et al., 2024) overlook the domain gap problem where training and testing data denoted as $\mathcal{D}^s$ and $\mathcal{D}^t$, respectively, follow different distributions due to uncontrolled variations in the acquisition setting (Cao et al., 2023; Carvalho et al., 2024). This domain shift induces, therefore, a significant drop in performance. To solve this issue, a handful of *Domain Generalization* (DG) methods for UAD have been proposed recently (Cohen et al., 2023; Carvalho et al., 2024; Cao et al., 2023). Cohen et al. (2023) propose a domain-disentanglement approach that removes predefined nuisance attributes (e.g., pose, lighting) from the source features using contrastive loss, preventing these factors from interfering with the anomaly task, improving the performance on unseen domains. However, without an actual target set, this method requires defining and labeling nuisance factors within the source dataset, which is challenging, as mentioned in their paper. In (Carvalho et al., 2024), multiple source domains are considered for learning domain-invariant features, thereby assuming the availability of diverse large-scale datasets, which is not always guaranteed. To avoid relying on multiple domains during training, a self-supervised strategy is adopted in (Cao et al., 2023). Nevertheless, the success of this approach heavily depends on the similarity between the augmented data and target samples. As a result, it necessitates tailoring augmentation techniques to unseen target datasets, if at all possible. Given its effectiveness, *Domain Adaptation* has also been explored to address the domain shift problem in AD (Cohen & Wolf, 2019; Li et al., 2021; Kumagai et al., 2019; Li et al., 2023; 2022). Those techniques usually adopt a few-shot adaptation paradigm by having access to a limited number of annotated target samples. While these methods offer innovative solutions for aligning source and target normal data, they still rely on costly annotations (Hermary et al., 2025) and are exposed to overfitting risks (Song et al., 2023). This emphasizes the need for a fully unsupervised domain adaptation for UAD. However, addressing this problem remains difficult due to the unsupervised nature of anomaly detection and domain adaptation, as discussed in the next.

## 3   THE TWO-FOLD UNSUPERVISED CURSE

Let us denote as $\mathcal{D}^s = \{(\mathbf{X}_i^s, y_i^s)\}_{i=1}^{N_s}$ a labeled dataset from a given domain called *source* formed by $N_s$ samples, where a sample $\mathbf{X}_i^s \in \mathbb{R}^{h \times w \times c}$ and its associated label $y_i^s \in \{0, 1\}$, $\forall i = \{1, ..., N_s\}$. Let $\mathcal{D}^t$ be a second unlabeled dataset from a different domain, i.e., *target*, denoted as $\mathcal{D}^t = \{\mathbf{X}_i^t\}_{i=1}^{N_t}$ and formed by $N_t$ samples where $\mathbf{X}_i^t \in \mathbb{R}^{h \times w \times c}$, $\forall i = \{1, ..., N_t\}$. In the following, we assume that $\mathcal{D}^t$ shares the same label space as $\mathcal{D}^s$ and that there exists a domain gap between $\mathcal{D}^s$ and $\mathcal{D}^t$. The goal of Unsupervised Domain Adaptation (UDA) for anomaly detection (whether formulated as a binary or one-class classification problem), is to learn a model $\zeta : \mathbb{R}^{h \times w \times c} \to \{0, 1\}$ using both $\mathcal{D}^s$ and $\mathcal{D}^t$ that generalizes to the target domain. In other words, it aims at learning a domain invariant feature extractor $f : \mathbb{R}^{h \times w \times c} \mapsto \mathcal{X}$ such that $\zeta = g \circ f$ with $g : \mathcal{X} \mapsto \{0, 1\}$ being the classifier and $\mathcal{X}$ the feature space given by $f$. This objective is achieved by minimizing the following adaptation upper bound (Ben-David et al., 2006),

$$\epsilon^t \leq \epsilon^s + d(f(\mathcal{D}^s), f(\mathcal{D}^t)) + \lambda \, , \tag{2}$$

where $\epsilon^t$ and $\epsilon^s$ are the expected classification errors on the target and source domains, respectively; $d(f(\mathcal{D}^s), f(\mathcal{D}^t))$ estimates the discrepancy between the feature distributions from the two domains, and $\lambda$ accounts for the joint error on source and target of an ideal detector.

While strategies for minimizing this upper bound are feasible in the context of binary or even multi-class classification (Kalluri et al., 2024; Singh et al., 2024; Wilson & Cook, 2020), the non-availability of anomalous data during training makes it difficult in the context of one-class classification, where $d(f(\mathcal{D}^s), f(\mathcal{D}^t))$ cannot be estimated. In fact, we can only use a subset $\mathcal{D}^{s,n} \subset \mathcal{D}^s$ formed by normal data for training. For that reason, existing works on domain adaptation for one-class anomaly detection (Cohen & Wolf, 2019; Li et al., 2023; Yang et al., 2023) revisit the formulation given in Eq equation 2 by slightly simplifying the problem. They pose it as a few-shot domain adaptation setting (instead of a fully unsupervised scenario). This means that they assume having access to a small labeled subset $\mathcal{D}^{t,n} \subset \mathcal{D}^t$ composed of normal samples only. As a result, they reformulate Eq equation 2 as,

$$\epsilon^{t,n} \leq \epsilon^{s,n} + d(f(\mathcal{D}^{s,n}), f(\mathcal{D}^{t,n})) + \lambda \, , \tag{3}$$

where $\epsilon^{s,n}$ and $\epsilon^{t,n}$ represent the source and target expected classification errors related to the normal class, respectively, since $\epsilon^s$ is not measurable in this context.

Nevertheless, in a fully unsupervised setup, we have access to $\mathcal{D}^t = \mathcal{D}^{t,a} \cup \mathcal{D}^{t,n}$ where $\mathcal{D}^{t,a}$ represents the subset of $\mathcal{D}^t$ formed by anomalies, without any prior information regarding the labels. Hence, directly aligning the feature distributions estimated from the source and target data by approximating $d(f(\mathcal{D}^{s,n}), f(\mathcal{D}^t))$ would lead to obtaining a classification boundary that is completely obsolete for target data, as shown in Figure 1 (b). We call this problem the *two-fold unsupervised curse* as it is a consequence of a lack of supervision: (1) in the task of anomaly detection, as it is formulated as a one-class problem where only normal source data are used; and (2) in the task of domain adaptation which is fully unsupervised where only an unlabeled target set is available. Given that the problem is ill-posed, it remains a significant challenge that has not been addressed in the existing UAD literature.

## 4   RARE ANOMALIES TO THE RESCUE

To tackle the two-fold unsupervised curse described in Section 3, we introduce a key assumption and the main hypothesis it entails for enabling UDA for one-class anomaly detection.

***Assumption (anomaly scarcity).*** *For an unlabeled target dataset $\mathcal{D}^t = \mathcal{D}^{t,n} \cup \mathcal{D}^{t,a}$, we assume that the number of anomalous samples is significantly smaller than the number of normal samples, i.e., $|\mathcal{D}^a| \ll |\mathcal{D}^n|$, with $|\cdot|$ refers to the cardinality.*

***Hypothesis (dominant cluster existence).*** *Considering a target unlabeled anomaly detection dataset $\mathcal{D}^t = \mathcal{D}^{t,n} \cup \mathcal{D}^{t,a}$ **under the anomaly scarcity assumption**, where $\mathcal{D}^{t,n}$ and $\mathcal{D}^{t,a}$ are respectively the normal and abnormal subsets, we hypothesize that there exists a feature extractor $\psi : \mathbb{R}^{h \times w \times c} \to \mathcal{X}$ that generates from $\mathcal{D}^t$ a compact dominant cluster $\mathcal{C} \in \mathcal{X}$ predominated by normal samples.*

*The anomaly scarcity assumption* often holds as it reflects most real-world scenarios where anomalies are rare compared to normal instances. Our main objective is therefore to find a feature extractor that

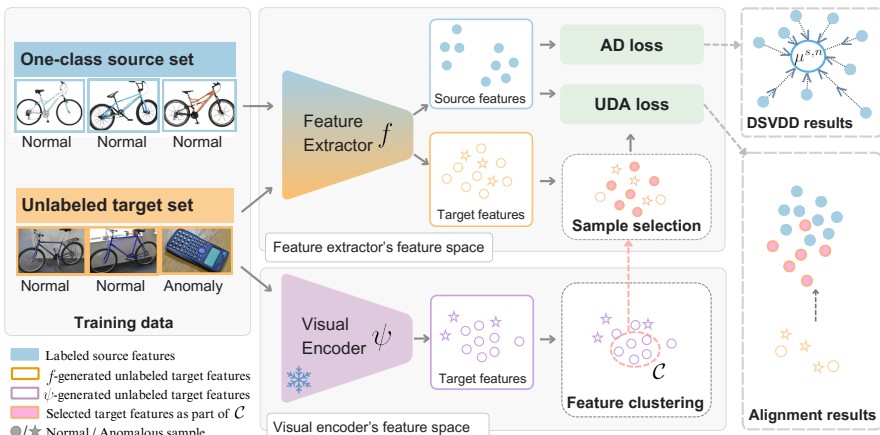

Figure 3. **Our Solution**: The top branch uses a trainable feature extractor with a DSVDD objective for one-class source data. The bottom branch extracts CLIP visual features, clusters them, identifies the dominant feature cluster and it corresponding features in the feature space of $f$ and align the latter with normal source features. ● are normals and ★ are anomalies.

verifies *the dominant cluster existence hypothesis*. We emphasize that this hypothesis is not granted and remains challenging. Nevertheless, it is a core component of the proposed method discussed in Section 5, as it enables the introduction of a novel paradigm to approach UDA for one-class UAD. The paradigm consists of the following steps: (1) finding a feature exactor $\psi$ that can generate a compact dominant cluster of features $\mathcal{C}$ corresponding to normal samples within an unlabeled target dataset $\tilde{\mathcal{D}}^t$, (2) identifying the subset of samples $\tilde{\mathcal{D}}^{t,n}$ corresponding to this cluster in the feature space of $\psi$, and (3) aligning the identified subset $\tilde{\mathcal{D}}^{t,n}$ with the source normal samples $\mathcal{D}^{s,n}$ in the feature space of the source feature extractor $f$. Formally, we revisit Eq (3) as follows,

$$\epsilon^{t,n} \leq \epsilon^{s,n} + d(f(\mathcal{D}^{s,n}), f(\tilde{\mathcal{D}}^{t,n})) + \lambda , \tag{4}$$

where $\tilde{\mathcal{D}}^{t,n} = \{\mathbf{X}_i^t \mid \psi(\mathbf{X}_i^t) \in \mathcal{C}\}$. Note that $\psi$ can be obtained by focusing on learning compact cross-domain features from which $\mathcal{C}$ can be identified through feature grouping and selection techniques such as clustering or filtering. As such, the proposed paradigm for UDA in one-class UAD lays the foundation for future research, where several technical choices can be explored at each stage.

## 5 METHODOLOGY

Building on the assumption and hypothesis formulated in Section 4, we present our methodology for introducing UDA to unsupervised visual *semantic* one-class anomaly detection, as one possible solution for tackling the two-fold unsupervised curse under this setting.

Our approach has two branches. The upper branch depicts a trainable backbone $f$ that learns from both source and target domain data. The source features are optimized using a Deep Support Vector Data Description (DSVDD) objective (Ruff et al., 2018). The lower branch focuses on visual feature extraction from the unlabeled target domain, through a frozen CLIP visual encoder (Radford et al., 2021), defined as the $\psi$ feature extractor. Clustering is applied to these visual features to estimate the dominant cluster $\mathcal{C}$. Samples identified within $\mathcal{C}$ in the $\psi$ visual encoder's representation space are then selected within the space of the feature extractor $f$ and then aligned with the normal source features. An algorithm is given in Section A.

**Training.** Specifically, given source and target image datasets $\mathcal{D}^{s,n}$ and $\mathcal{D}^t$, we apply DSVDD on the source data, enforcing feature compactness by minimizing the radius of a hypersphere to encapsulate the normal source representations. This is done by solving the following optimization problem,

$$\min_{\theta_f} \mathcal{L}_{AD} = \min_{\theta_f} \frac{1}{N_s} \sum_{i=1}^{N_s} \|f(\mathbf{X}_i^s) - \boldsymbol{\mu}^{s,n}\|_2^2, \forall \mathbf{X}_i^s \in \mathcal{D}^{s,n}, \tag{5}$$

where $\boldsymbol{\mu}^{s,n}$ is the mean of the source features. For clustering, we use a $K$-means algorithm. Note that $\psi$ can be $f$ itself in a self-training fashion or any frozen visual encoder such as CLIP (Radford

Table 1. Ten-run average and standard deviation of AUC (%) on the Office datasets.

### (a) OfficeHome (Venkateswara et al., 2017)

| Normal class | Source only DSVDD | Few-shot adaptation | | | | | | Unsup. adapt. Ours |
|---|---|---|---|---|---|---|---|---|
| | | BiOST | TSA | ILDR | IRAD | MsRA | Ours | |
| **Clip Art → Product (C → P)** | | | | | | | | |
| Bike | 97.48 | 43.00 | 69.10 | 89.90 | 90.30 | 94.30 | 98.34 | 85.71 |
| Calculat. | 83.47 | 69.00 | 72.20 | 84.90 | 82.20 | 98.70 | 97.76 | 97.70 |
| Drill | 81.57 | 66.40 | 66.20 | 75.30 | 73.00 | 84.50 | 74.19 | 96.64 |
| Hammer | 83.32 | 50.10 | 77.40 | 74.70 | 84.50 | 80.10 | 89.55 | 82.63 |
| Kettle | 87.74 | 63.00 | 63.10 | 77.50 | 75.80 | 85.50 | 94.08 | 89.16 |
| Knives | 78.09 | 48.80 | 51.90 | 55.20 | 63.90 | 64.40 | 79.25 | 76.63 |
| Pan | 74.00 | 57.70 | 63.70 | 72.20 | 76.00 | 80.50 | 93.08 | 91.07 |
| Paperclip | 53.04 | 27.40 | 74.70 | 76.70 | 67.40 | 79.70 | 71.18 | 67.98 |
| Scissors | 86.45 | 56.40 | 64.70 | 79.50 | 68.90 | 85.50 | 87.71 | 88.43 |
| Soda | 51.21 | 50.20 | 57.40 | 70.30 | 53.30 | 72.40 | 61.16 | 92.37 |
| Avg. | 77.64 | 53.20 | 66.04 | 75.82 | 73.53 | 82.56 | 84.63 | **86.83** |
| ±std | ±14.04 | ±11.65 | ±7.36 | ±8.81 | ±10.24 | ±9.33 | ±11.93 | ±8.66 |
| **Product → Clip Art (P → C)** | | | | | | | | |
| Bike | 82.55 | 52.70 | 65.80 | 83.10 | 85.70 | 86.60 | 82.06 | 92.99 |
| Calculat. | 62.82 | 65.20 | 63.40 | 87.20 | 79.20 | 91.90 | 91.59 | 89.88 |
| Drill | 71.81 | 47.00 | 57.10 | 63.90 | 71.20 | 73.50 | 70.58 | 77.54 |
| Hammer | 68.02 | 43.70 | 68.60 | 60.20 | 77.00 | 73.00 | 84.33 | 65.42 |
| Kettle | 71.85 | 47.70 | 61.50 | 68.80 | 70.00 | 73.40 | 75.38 | 78.19 |
| Knives | 57.22 | 63.10 | 57.50 | 65.30 | 70.30 | 73.10 | 77.74 | 71.99 |
| Pan | 71.44 | 49.30 | 63.50 | 69.30 | 72.80 | 80.00 | 83.72 | 82.46 |
| Paperclip | 26.19 | 45.10 | 49.90 | 69.70 | 61.80 | 69.00 | 67.05 | 55.93 |
| Scissors | 63.42 | 38.60 | 70.10 | 66.20 | 70.00 | 72.30 | 86.35 | 77.63 |
| Soda | 66.82 | 56.90 | 55.80 | 60.20 | 63.29 | 59.40 | 69.08 | 62.63 |
| Avg. | 64.21 | 50.93 | 61.32 | 69.39 | 72.13 | 75.22 | **78.79** | 75.47 |
| ±std | ±14.22 | ±8.11 | ±5.94 | ±8.55 | ±6.76 | ±8.62 | ±7.74 | ±11.13 |

### (b) Office31 (Saenko et al., 2010)

| Normal class | Source only DSVDD | Few-shot adaptation | | | | | | Unsup. adapt. Ours |
|---|---|---|---|---|---|---|---|---|
| | | BiOST | TSA | ILDR | IRAD | MsRA | Ours | |
| **Webcam → Amazon (W → A)** | | | | | | | | |
| Backpack | 86.48 | 59.90 | 76.30 | 91.90 | 90.20 | 95.20 | 95.40 | 97.62 |
| Bookcase | 35.77 | 56.60 | 59.60 | 78.40 | 82.20 | 84.50 | 76.25 | 91.16 |
| Bottle | 70.00 | 60.80 | 66.80 | 74.50 | 72.10 | 74.00 | 72.48 | 77.32 |
| Chair | 56.92 | 57.60 | 63.40 | 85.30 | 80.90 | 87.20 | 85.50 | 92.06 |
| Lamp | 82.26 | 50.50 | 60.90 | 72.60 | 67.50 | 70.00 | 82.38 | 81.50 |
| Headpho. | 88.91 | 57.60 | 75.90 | 88.90 | 81.60 | 92.20 | 92.53 | 95.06 |
| Keyboard | 79.83 | 58.20 | 69.90 | 88.30 | 93.20 | 95.40 | 95.40 | 93.36 |
| Laptop | 51.79 | 59.10 | 63.00 | 86.20 | 98.10 | 99.00 | 95.63 | 79.97 |
| Mouse | 83.95 | 65.80 | 53.40 | 84.90 | 79.60 | 89.90 | 96.65 | 92.97 |
| Pen | 48.54 | 68.50 | 69.10 | 75.50 | 71.40 | 73.90 | 72.72 | 71.20 |
| Avg. | 68.45 | 59.46 | 65.83 | 82.65 | 81.68 | 86.13 | 86.49 | **87.22** |
| ±std | ±17.84 | ±4.70 | ±6.86 | ±6.46 | ±9.37 | ±9.72 | ±9.44 | ±8.48 |
| **Amazon → Webcam (A → W)** | | | | | | | | |
| Backpack | 79.42 | 47.90 | 59.00 | 81.60 | 91.20 | 97.50 | 99.28 | 97.59 |
| Bookcase | 60.68 | 49.90 | 72.30 | 88.90 | 89.40 | 93.10 | 85.23 | 94.29 |
| Bottle | 40.94 | 66.00 | 69.80 | 86.90 | 95.30 | 96.20 | 93.65 | 94.95 |
| Chair | 71.66 | 67.00 | 66.20 | 76.10 | 90.30 | 90.10 | 93.67 | 99.08 |
| Lamp | 94.63 | 55.50 | 68.60 | 73.10 | 81.30 | 83.90 | 94.57 | 97.61 |
| Headpho. | 70.99 | 68.30 | 72.40 | 93.70 | 91.60 | 96.00 | 96.54 | 96.04 |
| Keyboard | 77.90 | 66.00 | 76.90 | 91.10 | 95.70 | 98.10 | 90.62 | 76.59 |
| Laptop | 91.61 | 62.10 | 72.20 | 85.70 | 97.10 | 98.20 | 94.32 | 97.67 |
| Mouse | 72.17 | 69.10 | 69.40 | 82.20 | 85.40 | 86.50 | 96.35 | 81.41 |
| Pen | 44.26 | 79.10 | 86.10 | 97.60 | 98.90 | 99.60 | 97.09 | 99.99 |
| Avg. | 70.43 | 63.09 | 71.29 | 85.69 | 91.62 | 93.92 | **94.13** | 93.52 |
| ±std | ±16.81 | ±9.03 | ±6.66 | ±7.26 | ±5.15 | ±5.11 | ±3.72 | ±7.52 |

et al., 2021) or DINO-v2 (Oquab et al., 2023). The dominant cluster is identified as,

$$\mathcal{C} = \arg\max_{\mathcal{C}_k} |\mathcal{C}_k| \text{ for } k \in \{1, ..., K\}, \tag{6}$$

where $|\mathcal{C}_k|$ is the size of the $k$-th cluster $\mathcal{C}_k$, and $K$ is a hyperparameter defining the number of expected components in the space of $\psi(\mathcal{D}^t)$. When clustering is applied to $f(\mathcal{D}^t)$, the selected features for alignment are $\tilde{\mathcal{D}}^{t,n} = \mathcal{C}$. When clustering is applied to $\psi(\mathcal{D}^t)$, the selected samples are:

$$\tilde{\mathcal{D}}^{t,n} = \{f(\mathbf{X}_i^t) \mid \psi(\mathbf{X}_i^t) \in \mathcal{C}\} \; \forall \mathbf{X}_i^t \in \mathcal{D}^t \tag{7}$$

Alignment between source and target features is achieved using a contrastive strategy, where UDA loss is computed as:

$$\mathcal{L}_{UDA} = \frac{1}{N_s \times |\tilde{\mathcal{D}}^{t,n}|} \sum_{i=1}^{N_s} \sum_{j=1}^{|\tilde{\mathcal{D}}^{t,n}|} \ell_{i,j}, \text{ with } \ell_{i,j} = -\log \frac{\exp\left(\frac{1}{\tau} \cdot \text{sim}\left(f(\mathbf{X}_i^s), f(\mathbf{X}_j^t)\right)\right)}{\sum_{p=1}^{N_t} \mathbb{1}_{[\mathbf{X}_p^t \notin \tilde{\mathcal{D}}^{t,n}]} \exp\left(\frac{1}{\tau} \cdot \text{sim}\left(f(\mathbf{X}_i^s), f(\mathbf{X}_p^t)\right)\right)} \tag{8}$$

where $\text{sim}(\cdot, \cdot)$ denotes the cosine similarity, and $\tau$ is the temperature. Finally, the overall loss is:

$$\mathcal{L} = \lambda_1 \, \mathcal{L}_{AD} + \lambda_2 \, \mathcal{L}_{UDA}, \tag{9}$$

where $\lambda_1$ and $\lambda_2$ are hyperparameters for $\mathcal{L}_{AD}$ and $\mathcal{L}_{UDA}$.

**Inference.** Note that the visual encoder $\psi$ is discarded at inference and only the feature extractor $f$ is used to determine whether the input data is anomalous by calculating whether it falls inside or outside the hypersphere estimated by the DSVDD model. Our algorithm is given in Section A.

## 6 EXPERIMENTAL RESULTS

### 6.1 EXPERIMENTAL SETTING

This section describes the datasets, the baselines used and the implementation details of our experiments. We report the performance using Area Under the ROC Curve (AUC) using **bold** and underline for the best and second performances, respectively. Section B.1 and Section B.2 provide further details on each subsection, and additional experiments.

**Datasets.** We evaluate our approach on four standard UDA benchmark datasets, **Office-Home** (Saenko et al., 2010), **Office31** (Saenko et al., 2010), **VisDA** (Peng et al., 2017), and **PACS** (Li et al., 2017). The types of domain shift of each dataset is described in Section A. For the AD task, we adopt a standard one-vs-all protocol, since we focus specifically on the one-class setting, where a single class is available as normal and the remaining are anomalies. We adopt the experimental protocol of previous DA works (Li et al., 2023; Yang et al., 2023; Cohen & Wolf, 2019) to allow

for a fair comparison –that is, we show results on ten classes from the ClipArt and Product domains for Office-Home, ten classes from Webcam and Amazon for Office31, and twelve classes from the domains of Computer Aided Designs (CAD) (synthetic objects) and real object photos of VisDA. On PACS, like (Cao et al., 2023), we use Photo as source domain and the remaining domains as targets. **Baselines.** As no other works on UDA for visual semantic UAD were previously introduced, we compare our method with several few-shot adaptation SoA approaches. Specifically, we consider **BiOST** (Cohen & Wolf, 2019) which is a one-shot approach, **TSA** (Li et al., 2021), **ILDR** (Kumagai et al., 2019), **IRAD** (Yang et al., 2023), and **MsRA** (Li et al., 2023) that are few-shot adaptation methods. Furthermore, we introduce our few-shot adaptation variant (**Ours-Few-shot**), which augments the target domain with normal and pseudo-anomalous samples similar to (Ge et al., 2021). This augmentation yields semantically positive and negative pairs (Ge et al., 2021), useful for the contrastive alignment strategy described in Section 5. Further details are given in the Section A.

**Implementation details.** In all experiments, the source set has only one-class normal data, while the unlabeled target set includes mostly normals with 10% randomly sampled anomalies. Training uses SGD with a cosine-annealing scheduler, learning rate of $10^{-3}$, weight decay of $5 \times 10^{-7}$, batch size 256 and $\lambda_1$ and $\lambda_2$ are set to 1. CLIP-ViT-B32 is the frozen visual encoder $\psi$ for feature clustering. Contrastive loss temperature $\tau$ is 0.07. To align with the setting of the baselines (Li et al., 2023; Yang et al., 2023; Cohen & Wolf, 2019; Li et al., 2021), ResNet50 is the trainable backbone $f$, initialized on ImageNet (Deng et al., 2009). K-means clustering (Hartigan & Wong, 1979) uses 2, 10, and 5 components for Office, VisDA, and PACS, respectively. Like the baselines, the few-shot adaptation settings use 10 (Office, PACS) and 100 shots (VisDA) **labeled as normal**, respectively.

## 6.2 COMPARISON AGAINST STATE-OF-THE-ART.

Our method outperforms previous SoA on all benchmarks of our evaluation, as shown in Table 1 and Table 2. More specifically, our fully unsupervised variant importantly improves upon previous few-shot adaptation SoA on C → P and W → A of the Office-Home and Office31 datasets. In addition, we observe an improvement of over 10% in the VisDA dataset with the fully unsupervised methodology over previous few-shot adaptation approaches, despite being challenged by the two-fold unsupervised curse. These results highlight the relevance of the proposed method, even in the presence of a large domain gap, as in the case of synthetic CAD images and real-world photos.

In the P → C and A → W adaptation of the Office datasets, our few-shot adaptation variant also registers SoA performance, closely followed by our model trained under the fully unsupervised setting. These results highlight the flexibility of our framework, which can leverage minimal labeled target data when available but remains highly effective in a fully unsupervised setup.

Table 2. AUC (%) on the target domain of our UDA anomaly detector on VisDA compared **with various adaptation paradigms** (zero-shot, i.e., pretrained Visual encoders, few-shot, and supervised, i.e., Oracle).

| Normal class | w/o adaptation | | | w/ adaptation | | | | |
|---|---|---|---|---|---|---|---|---|
| | Zero-shot | | Source finetuned DSVDD | Few-shot | | | Unsup. **Ours** | Super-vised |
| | R50 | CLIP | | BiOST | MsRA | Ours | | |
| **CAD → Real** | | | | | | | | |
| Aero. | 41.05 | 74.97 | 67.71 | 36.80 | 81.56 | 81.55 | 84.86 | 90.91 |
| Bicycle | 67.35 | 90.28 | 65.12 | 59.20 | 68.45 | 74.58 | 81.45 | 81.73 |
| Bus | 28.58 | 42.27 | 66.01 | 47.90 | 68.12 | 72.26 | 82.17 | 72.16 |
| Car | 32.48 | 64.16 | 78.65 | 53.80 | 69.44 | 82.78 | 62.76 | 68.42 |
| Horse | 68.81 | 75.48 | 67.24 | 58.00 | 68.77 | 80.17 | 83.52 | 88.70 |
| Knife | 67.78 | 95.28 | 62.43 | 54.10 | 70.39 | 71.52 | 68.82 | 78.90 |
| Motor. | 60.07 | 82.25 | 69.45 | 58.10 | 65.64 | 80.16 | 91.15 | 83.46 |
| Person | 71.69 | 56.26 | 42.11 | 58.70 | 59.18 | 51.24 | 69.68 | 85.19 |
| Plant | 62.47 | 89.65 | 57.77 | 42.10 | 65.81 | 71.46 | 70.58 | 82.63 |
| Skate. | 85.00 | 91.52 | 60.70 | 41.60 | 61.30 | 63.17 | 83.71 | 83.73 |
| Train | 30.13 | 57.74 | 54.75 | 52.40 | 69.73 | 60.62 | 69.98 | 85.11 |
| Truck | 26.05 | 45.08 | 62.08 | 43.10 | 59.05 | 73.67 | 57.84 | 78.91 |
| Avg. | 53.45 | 72.08 | 62.84 | 50.48 | 67.28 | 71.93 | 75.54 | **81.65** |
| ±std | ±19.57 | ±17.83 | ±8.55 | ±7.55 | ±5.79 | ±9.08 | ±9.80 | ±6.11 |

variant also registers SoA performance, closely followed by our model trained under the fully unsupervised setting. These results highlight the flexibility of our framework, which can leverage minimal labeled target data when available but remains highly effective in a fully unsupervised setup.

Furthermore, we compare the performance of our model to two pretrained visual encoders, namely ResNet50 and CLIP-ViT-B32 in Table 2. While the CLIP-ViT-B32 architecture achieves an average AUC of 72.08%, our unsupervised method (75.54%) still outperforms it on the VisDA dataset. In contrast, the ResNet50 shows a significantly lower performance, with an average AUC of only 53.45%. These results demonstrate that despite their strong performance, pretrained visual encoders are not specifically tailored for the domain adaptation task; thus, they remain vulnerable to domain shift. Therefore, training domain adaptation-specialized models is still necessary to effectively bridge the gap between two domains.

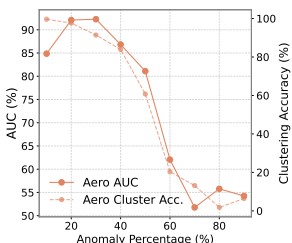

Figure 4. Validity of the anomaly scarcity assumption.

## 6.3 ADDITIONAL EXPERIMENTS

Unless stated otherwise, all the following experiments are performed on VisDA (Peng et al., 2017). Additional results on different components of our method are given in Section B.1.

**Anomaly scarcity assumption.** To evaluate the impact of anomaly scarcity, we vary the anomaly ratio in the unlabeled target set from 10% to 90% and report our method's performance alongside clustering accuracy in Figure 4 for the Aeroplane class from VisDA. The results indicate a strong correlation between AUC performance and clustering accuracy. As the anomaly proportion increases, the AUC gradually degrades, with a drastic drop beyond 50%, where the dominant cluster assumption no longer holds. This is further evidenced by a significant decrease in the clustering accuracy.

**Few-shot versus unsupervised adaptation paradigms.** The results presented in Table 2 compare pretrained visual encoders and source-only detectors with different adaptation paradigms, i.e., few-shot, unsupervised, and supervised (oracle). The source-only finetuned model improves slightly over the pretrained ResNet50 visual encoder but still has lower performance than the adaptation approaches, achieving an average AUC of 62.84%. Among the few-shot methods, our few-shot adaptation variant outperforms BiOST and MsRA, achieving the highest AUC of 71.93%, which is comparable to the performance of a pretrained CLIP-ViT-B32 visual encoder. However, our unsupervised adaptation method surpasses all these models, with an average AUC of 75.54% indicating its ability to effectively mitigate domain gaps without relying on labeled target data. This can be explained by the fact that after clustering, our model has access to more representative normal target data than few-shot models, hence better generalizing to the target normal class. Furthermore, the Oracle, which has access to the target labels, achieves the highest performance (81.65%). The small gap between our unsupervised method and the oracle demonstrates the effectiveness of our approach even without supervision.

**Ablation on the framework components.** Table 3 provides the results obtained when each component, namely the use of a adaptation loss, the dominant cluster identification through clustering, the use of an auxiliary visual encoder $\psi(\mathcal{D}^t)$ or the trainable features $f(\mathcal{D}^t)$. The results show that without adaptation, a model trained only on

Table 3. Method components ablation.

| w/ Adaptation | w/ Clustering | w/ CLIP $\psi$ | AUC (%) |
|:---:|:---:|:---:|:---:|
| ✗ | ✗ | ✗ | 62.84±8.55 |
| ✓ | ✗ | ✗ | 64.33±6.42 |
| ✓ | ✓ | ✗ | 68.47±8.30 |
| ✓ | ✓ | ✓ | **75.54±9.80** |

source data generalizes poorly to the target domain with only 62.84%. Direct adaptation of the source and the unlabeled target without clustering leads to inconsistent results, indicating low generalization capabilities to the target domain. Introducing clustering results in a significant performance boost. This can be seen when clustering is applied to the original representations of the feature extractor, as the performance improves by +5.63%, highlighting the importance of identifying the dominant cluster prior to alignment. Note that our method still outperforms the best few-shot adaptation baseline MsRA (68.47% vs 67.28%) with just clustering and alignment (i.e., **w/o CLIP** where $\phi$ is self-trained) across all the VisDA classes. Finally, the best results are achieved when all components are combined. This setup boosts the average AUC to 75.54% on all VisDA classes. The substantial performance gains can be attributed to CLIP's rich visual features, which, together with clustering and alignment, help achieve a more robust anomaly detector capable of better handling domain shift. This remains valid when $\phi$ and $\psi$ are both CLIP (See Table S9 in the Appendix).

**Clustering methods.** We compare different clustering techniques on three UDA benchmarks in Table 5. The first observation we make is that any type of clustering improves the performance. K-means and GMM have comparable results, without one clearly and consistently outperforming the other across datasets and adaptation directions. Meanshift clustering offers a performance increase compared to source-only models. However, its performance remains lower than that of the other clustering methods. In our experiments, we chose K-means clustering as it achieves comparable performance to GMM while requiring fewer parameters and simpler optimization. We further investigate the optimal number of K-Means components, as shown in Figure 5. The figure indicates that using 8 to 10 components yields

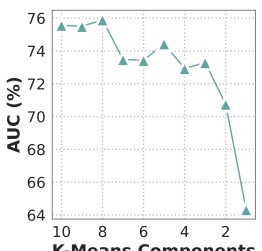

Figure 5. K-Means components variation on VisDA.

the highest performance, with an AUC of approximately 75-76%. Decreasing the number of components would gradually degrade the performance. This suggests that a lower number of clusters may not capture the characteristics of the majority class, leading to inaccurate clustering and thus negatively impacting the generalization of the anomaly detection model across domains. Similar

to (Reiss et al., 2021; Cohen et al., 2023), it uses a kNN density estimator to detect anomalies. Our results suggest that both methods benefit from the adaptation, as a consistent average improvement of +12.7% and +8.23% is seen across all 12 classes of VisDA.

**Beyond DSVDD by using other AD objectives.** To assess whether our alignment approach applies to other unsupervised AD methods, we replace DSVDD with Mean-shifted Contrastive loss (MSC) (Reiss & Hoshen, 2023) in Table 4. MSC adapts contrastive loss to the one-class setting by shifting augmented representations of the normal samples toward the mean of pretrained normal features, preserving their compactness. It can be seen that our unsupervised adaptation improves the performance on average by +8.23 on the VisDA dataset.

**Alignment strategies.** By comparing several alignment

Table 4. Our UDA method on anomaly detection methods (Ruff et al., 2018; Reiss & Hoshen, 2023). ZS and Src mean Zero-shot and Source only.

| $f$: ResNet50 + $\psi$: CLIP-ViT-B32 | | | | | |
|---|---|---|---|---|---|
| **DSVDD** | | | **MSC** | | |
| **ZS** | **Src** | **UDA** | **ZS** | **Src** | **UDA** |
| 53.45 | 62.84 | 75.54 (**+12.7**↑) | 74.39 | 72.87 | 81.10 (**+8.23**↑) |

strategies in Table 6, we observe that any alignment strategy, in general, improves the performance consistently for all adaptation benchmarks. Contrastive alignment consistently outperforms other adaptation losses, including statistical (MMD) (Gretton et al., 2006) and adversarial (GRL) (Ganin & Lempitsky, 2015) strategies.

Table 5. Clustering ablation. K=10 for GMM and K-means on VisDA and K=2 for other datasets. For kNN, k=2 for VisDA and k=1 for the rest.

| Dataset | w/o Adapt. (Src Only) | w/ Adaptation | | | |
|---|---|---|---|---|---|
| | | **KMeans** | **GMM** | **MeanShift** | **kNN** |
| VisDA | 62.84±08.55 | **75.54**±10.23 | 72.24±08.81 | 74.14±07.44 | 71.65±06.68 |
| A→W | 72.57±18.69 | 94.82±07.52 | **96.45**±05.43 | 87.32±12.53 | 82.73±10.96 |
| W→A | 67.70±18.32 | **87.72**±12.63 | 87.00±13.60 | 86.68±08.53 | 83.63±06.04 |
| C→P | 77.31±15.13 | 90.54±14.06 | **90.85**±11.18 | 85.95±10.94 | 78.67±15.19 |
| P→C | 63.88±15.60 | 70.92±11.36 | **76.15**±13.32 | 73.50±13.49 | 71.38±11.71 |

Table 6. AUC (%) using different domain adaptation losses ((Ganin & Lempitsky, 2015; Gretton et al., 2006)) on the considered datasets.

| Dataset | w/o Adapt. (Src Only) | w/ Adaptation | | |
|---|---|---|---|---|
| | | **GRL** | **MMD** | **Contrastive** |
| VisDA | 62.84±08.55 | 71.84±10.89 | 73.12±10.36 | **75.54**±9.80 |
| A→W | 72.57±18.69 | 90.43±10.68 | 86.86±12.97 | **94.82**±07.52 |
| W→A | 67.70±18.32 | 83.49±11.04 | 83.92±09.90 | **87.72**±12.63 |
| C→P | 77.31±15.13 | 83.40±12.35 | 82.10±12.16 | **90.54**±14.06 |
| P→C | 63.88±15.60 | 66.78±15.02 | 67.00±14.41 | **70.92**±11.35 |

## 7 LIMITATIONS AND FUTURE WORK

Our method, presented in Section 5, is one possible solution for addressing the problem of UDA for UAD. However, it is worth noting that it was tested in the context of *semantic anomaly detection* (Sträter et al., 2024), adopting a one-vs-all protocol, to facilitate the comparison with the closest baselines, namely (Li et al., 2023; Yang et al., 2023). These methods typically require the use of global features in contrast to standard anomaly detection, where fine-grained representations are usually targeted. For that reason, our method focuses mostly on global representations, while local features would be conceptually more suitable for fine-grained anomaly detection. In future works, we aim to extend our study to fine-grained anomaly detection by exploiting more relevant local representations, such as industrial and medical UAD.

## 8 CONCLUSION

This work is the first to address unsupervised domain adaptation (UDA) for one-class-based unsupervised anomaly detection (UAD), subject to what we refer to as the two-fold unsupervised curse. To address this ill-posed problem, an inherent property of anomalies, namely, their scarcity, is leveraged. This characteristic allows utilizing clustering, –as one possible solution– for identifying a dominant cluster within the unlabeled target set. Assuming this cluster to be predominantly composed of normal data, a contrastive alignment strategy is then used to align its features with the normal source representations. Extensive experiments on standard UDA benchmarks demonstrate that the proposed method effectively mitigates the domain gap and enhances anomaly detection performance across different domains, outperforming other supervised adaptation approaches without requiring target annotations. Finding the optimal feature extractor remains an open research question. In future work, we intend to further explore compact representations across domains to improve the proposed domain adaptation framework.

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

# WHEN UNSUPERVISED DOMAIN ADAPTATION MEETS ONE-CLASS ANOMALY DETECTION: ADDRESSING THE TWO-FOLD UNSUPERVISED CURSE BY LEVERAGING ANOMALY SCARCITY

## – Appendix –

## OVERVIEW

In this supplemental material, we provide additional details about the experimental setup and performance evaluation. More specifically, Section A elaborates on our method's algorithm, the baselines, datasets, evaluation metrics, and overall implementation details, while Section B discusses additional quantitative and qualitative results, supplementing the main results presented in Section 6.

## A  EXPERIMENTAL SETTING DETAILS

---

**Algorithm 1** Training procedure of UDA for UAD

---

**Require:** $\mathcal{D}^{s,n}, \mathcal{D}^t$ source and target datasets, $K$ cluster components,
   $N_{\text{epochs}}, N_{\text{iter}}, \textbf{trainable } f, \textbf{frozen } \psi$

1: $s\_\text{feats\_center} = \texttt{mean}(f(\mathcal{D}^{s,n}))$        ▷ Set src feats center for DSVDD
2: **for** 1 to $N_{\text{epochs}}$ **do**
3:     **for** $1 \leftarrow N_{\text{iter}}$ **do**
4:         $s\_\text{feats}, t\_\text{feats} \leftarrow f(s\_\text{batch}, t\_\text{batch})$
5:         $t\_\text{CLIP\_feats} \leftarrow \psi(t\_\text{batch})$
6:         $t\_\text{clusters} \leftarrow \texttt{KMeans}(t\_\text{CLIP\_feats}, K)$
7:         $\text{indices} \leftarrow \texttt{dom\_cluster\_indices}(t\_\text{clusters})$        ▷ Eq (6,7)
8:         $t\_\text{dominant\_feats} \leftarrow t\_\text{feats}[\text{indices}]$
9:         $t\_\text{non\_dominant\_feats} \leftarrow t\_\text{feats}[\neg\text{indices}]$
10:        $\text{pos\_pairs} \leftarrow \texttt{pair}(s\_\text{feats}, t\_\text{dominant\_feats})$
11:        $\text{neg\_pairs} \leftarrow \texttt{pair}(s\_\text{feats}, t\_\text{non\_dominant\_feats})$
12:        $\mathcal{L}_{\text{UDA}} \leftarrow \texttt{contrast}(\text{pos\_pairs}, \text{neg\_pairs})$        ▷ Eq (8)
13:        $\mathcal{L}_{\text{UAD}} \leftarrow \texttt{DSVDD}(s\_\text{feats}, s\_\text{feats\_center})$        ▷ Eq (5)
14:        $\mathcal{L} \leftarrow \lambda_1 \mathcal{L}_{\text{UAD}} + \lambda_2 \mathcal{L}_{\text{UDA}}$        ▷ Eq (9)
15:        $\mathcal{L}.\texttt{backward}()$
16:        $\theta_f \leftarrow \texttt{update}(\theta_f)$
17:    **end for**
18: **end for**

---

**Algorithm.**  For each pair of source and target batches, our algorithm: (1) extracts source and target features using $f$ (ResNet50) and frozen visual target features using $\psi$ (CLIP-ViT-B32), (2) applies K-means to the frozen CLIP target features, (3) identifies the largest cluster and retrieves the indices of its samples within the original batch of target features, (4) uses those indices to select the corresponding target features extracted from ResNet50 using index-based selection, (5) defines accordingly positive and negative pairs, (6) computes the DSVDD loss using only source ResNet50 features, (7) computes contrastive loss with the positive and negative pairs as shown in Figure S7, and (8) backpropagates only through ResNet50 while CLIP remains frozen. At inference, only the distance of the target domain features from the source features center (line 1 in the algorithm) is used to assess whether those features are normal or anomalous.

**Baselines.**  We compare our method with the following few-shot domain adaptation for unsupervised anomaly detection methods, namely:
**BiOST** (Cohen & Wolf, 2019): is an unsupervised one-shot domain translation approach proposed for

learning a bidirectional mapping to translate the style of an image from one visual domain to another. It trains two autoencoders for each domain to match the input samples and their corresponding latent representations across domains using cycle-consistency losses. Given its autoencoder-based architecture, the reconstruction loss is used as an anomaly score to identify anomalies in the target domain. The source code[1] was used for our experiments. However, no license was provided for this work.

**TSA** (Li et al., 2021): is a classifier adaptation method that complements the source domain features with target domain semantics-aware augmentations obtained by probabilistic estimation of the target feature semantics. This is achieved by modeling the target domain distribution with a multivariate normal distribution. In the context of anomaly detection, an Isolation Forest (IF) model (Liu et al., 2008) trained on source-augmented features is used, making it a two-stage approach.

**ILDR** (Kumagai et al., 2019): estimates a latent domain vector from normal target samples to capture domain-specific knowledge and models it with a multivariate Gaussian distribution. It enables training-free knowledge transfer from the source domain to the target domain and infers a reconstruction-based anomaly detector from the corresponding target latent vector.

**IRAD** (Yang et al., 2023): is a two-stage approach that adversarially trains a shared encoder jointly with a source-specific encoder and a generator to extract common features of the source and target domains. A discriminator, together with a cycle-consistency loss, ensures that the extracted features capture shared information while excluding source-specific details. In the second stage, an IF (Liu et al., 2008) model is trained on the target-domain features.

**MsRA** (Li et al., 2023) is a single-stage approach. It introduces a feature extractor with a multi-spectral fusion module to process features in different frequency bands. This helps mitigate the information loss from limited target data. Adversarial training enables learning domain-invariant representations, while a center loss enforces compactness on normal data features. The anomaly score is calculated based on the distance from a sample to the prototype of normal class features. This design enables end-to-end training and inference for anomaly detection. The official source code[2] was used in our experiments. However, no license was provided for this work.

**Ours-Few-shot**: we propose a simple few-shot approach that follows the same protocol as MsRA (Li et al., 2023). Similar to (Kumagai et al., 2019; Li et al., 2021), it relies on data augmentation techniques such as the one proposed in (Ge et al., 2021) to generate semantic pseudo-anomalous samples. Positive pairs are then built between the source and normal target data, while negative pairs are constructed using the source and pseudo-anomalous instances. The contrastive loss ensures that positive pairs are similar while negative pairs are dissimilar in the feature space. Examples of augmented images and source-target pairs are depicted in Figure S7. Those design choices were made to accommodate the semantic anomaly detection protocol and enable comparison with baselines using the same protocol.

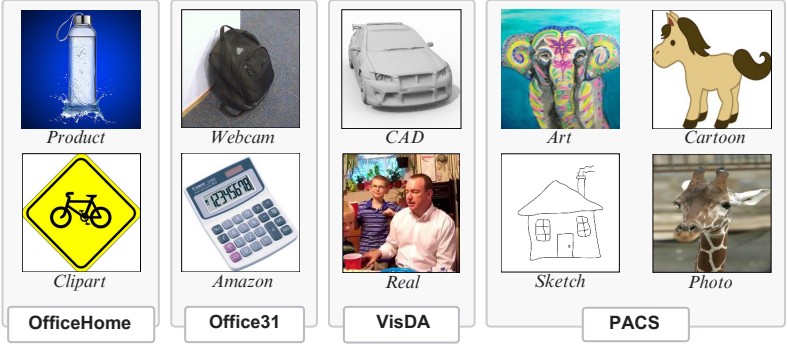

Figure S6. Examples of the selected domains from the datasets used in our paper. Each dataset includes multiple domains: OfficeHome (Venkateswara et al., 2017) (*Product*, *Clipart*), Office31 (Saenko et al., 2010) (*Amazon*, *Webcam*), VisDA (Peng et al., 2017) (*CAD*, *Real*), and PACS (Li et al., 2017) (*Art*, *Cartoon*, *Sketch*, *Photo*).

**Datasets.** Figure S6 gives examples of each domain used in our experiments for each dataset. Particularly, for the office datasets, we follow the same protocol as (Li et al., 2023), and for PACS (Li

---

[1] https://github.com/tomercohen11/BiOST
[2] https://github.com/BIT-DA/MsRA

Figure S7. Process for building the positive and negative pairs for the contrastive loss computation in the **Ours-Few-shot** variant. **(a)** The normal few samples of the target domain are augmented using a 4×4 jigsaw puzzle augmentation as proposed by (Ge et al., 2021), yielding semantically negative images that can be considered as pseudo-anomalies. **(b)** Then, the source data is paired with the normal target data (respectively with the pseudo-anomalous target data) to form positive pairs (respectively negative pairs). Examples are shown from PACS (Li et al., 2017) on the *Photo* and *Cartoon* domains.

et al., 2017), we also consider similar adaptation directions as (Cao et al., 2023). It is worth noting that these datasets have limited numbers of samples, especially under the one-versus-all one-class protocol. For example, in Office31 (Saenko et al., 2010), the average source set size and train target size across all classes and adaptation directions are 59 and 19 samples, respectively. Similarly, OfficeHome (Venkateswara et al., 2017) averages 89 source samples and 58 target samples. In contrast, VisDA (Peng et al., 2017), on average, has 12700 samples in the source set and 5077 target train samples with more challenging examples in the *Real* domain, as shown in Figure S6. This motivated us to focus mainly on VisDA (Peng et al., 2017) in our experiments, since it has the most data, especially under the one-class setting.

Table S7. Overview of datasets used, with download links and licensing information. Custom license is what `https://paperswithcode.com/datasets` states, when no clear license is mentioned on the official webpages.

| Dataset | Link | License |
|---|---|---|
| VisDA-2017 (Peng et al., 2017) | `https://ai.bu.edu/visda-2017/` | Custom license, but no explicit mention; created for a research challenge |
| Office-31 (Saenko et al., 2010) | `https://faculty.cc.gatech.edu/~judy/domainadapt/` | No explicit license provided; created for open academic use. |
| Office-Home (Venkateswara et al., 2017) | `https://www.hemanthdv.org/OfficeHome-Dataset/` | Custom license permitting non-commercial research & educational use. |
| PACS (Li et al., 2017) | `https://domaingeneralization.github.io/#data` | License not explicitly stated. Accompanying code has MIT License. |

**Evaluation metrics.** For assessing the anomaly detection performance on the target set, the Area Under the ROC Curve (**AUC**) is reported similarly to prior works (Cao et al., 2023; Li et al., 2023; Yang et al., 2023) as well as the average and standard deviation across all the classes of each dataset. Additional metrics, such as the Balanced Accuracy (**B.acc.**), Precision (**P.**), and Recall (**R.**), are provided in Section B.1.

Figure S8. Process of computing the source and target feature similarity matrix used for calculating the contrastive loss. The matrix is computed for all $N_s$ source and $N_t$ target samples. The contrastive loss ensures that the cosine similarity ($sim(.,.)$) between the dominant cluster $\mathcal{C}$ features and the source representations is then maximized (Pink and blue pairs). At the same time, it is minimized between the source and non-dominant cluster features (orange and blue pairs).

**Implementation details.** In all experiments, the source set contains only "normal" one-class samples, while the unlabeled target set includes both normal and anomalous data from various classes. In general, the target set is set to be half the size of the source set, which is mostly due to the actual sizes of the considered datasets. We assume that anomalies are rare in the target set and cover about 10% of the size of the normal data in the target domain[3]. Those anomalies are selected randomly from other target domain classes. For training, we use an SGD optimizer in all experiments with a cosine-annealing scheduler (Loshchilov & Hutter, 2016), a learning rate of $10^{-3}$, a weight decay of $5 \times 10^{-7}$, and a batch size of 256 samples. Unless specified otherwise, the frozen visual encoder used before feature clustering is CLIP-ViT-B32 (Radford et al., 2021; Dosovitskiy et al., 2020), selected for its rich semantic representations learned through extensive pre-training. For the Office datasets, we use a ResNet18 (He et al., 2016) as the trainable feature extractor backbone initialized with the ImageNet (Deng et al., 2009) weights, with 50 epochs and two-components K-means clustering (Hartigan & Wong, 1979). For the VisDA (Peng et al., 2017), we train a ResNet50 (He et al., 2016) backbone initialized with ImageNet (Deng et al., 2009) weights as the trainable feature extractor, with 30 epochs and ten-component K-means clustering (Hartigan & Wong, 1979). For the PACS (Li et al., 2017) dataset, we also use an ImageNet-pre-trained ResNet50 (He et al., 2016; Deng et al., 2009) backbone trained for 50 epochs and with a five-component K-means clustering (Hartigan & Wong, 1979). For the few-shot experiments, we follow the example of (Li et al., 2023), considering 10 shots for the Office and the PACS (Li et al., 2017) datasets and 100 shots for the VisDA dataset (Peng et al., 2017). Additionally, since contrastive loss requires negative pairs, and since we only have normal samples in the few-shot setting (i.e., positive pairs only), we build non-semantic negative pairs following the approach in (Ge et al., 2021). The scales of the anomaly and adaptation losses, $\lambda_1$ and $\lambda_2$ are both set to 1. The contrastive loss temperature $\tau$ is set to 0.07. Finally, all experiments were conducted on a 128-core Linux-based computer with an NVIDIA RTX A6000 GPU of 48 GB VRAM.

**Contrastive alignment details.** Similar to (Radford et al., 2021), we use a contrastive loss to maximize the similarity between the source and target samples, which, in turn, aligns the most representative samples of the target domain with the source domain. As depicted in Figure S8, the computed similarity matrix maximizes the similarity of the target samples belonging to the dominant cluster $\mathcal{C}$. In contrast, the instances outside of $\mathcal{C}$ have their similarity with the source samples minimized. Specifically, this is done by using a cross-entropy loss over those similarity scores. Note that in our few-shot variant, the similarity matrix is built in a similar fashion, using the labels of the normal target and the generated pseudo-anomalies.

---

[3]The exact protocol files will be released with the source code.

# B ADDITIONAL RESULTS

The results presented in the main paper in Section 6.2 and Section 6.3 focused on analyzing our method against SoA Domain Adaptation (DA) for UAD methods, ablating the framework components and varying the alignment strategies and paradigms. Herein, we complement those results by presenting additional metrics results in Table S11, comparing the contribution of various Pretrained Visual Encoders in Figure S9b and Table S10, and providing a sensitivity analysis in Figure S9a. Finally, we give qualitative results depicting the distributions of anomaly scores in Figure S10.

## B.1 QUANTITATIVE RESULTS

Table S8. AUC (%) of Domain Generalization (DG) for anomaly detection, trained **ONLY** on the source domain Photo (Ph.) and tested on unseen domains. DA means Domain Adaptation.

| Adapt. type | Method | Source domain: Photo → | | | Avg. ± std |
|---|---|---|---|---|---|
| | | Art | Cartoon | Sketch | |
| None | Source only | 64.06 | 64.08 | 57.35 | 61.83±3.17 |
| DG | GNL (Cao et al., 2023) | 65.62 | 67.96 | 62.39 | 65.32±2.28 |
| DA | MsRA (Li et al., 2023) (Few-shot) | **71.43** | 69.89 | 61.87 | 67.73±4.19 |
| | Ours (Unsup.) | 67.20 | **75.35** | **74.04** | **72.20**±3.57 |

**Comparison against domain generalization methods.** Table S8 compares the results of GNL (Cao et al., 2023) with DA methods on the PACS dataset (Li et al., 2017), with Photo as the source and Art, Cartoon, and Sketch as the target domains. It can be seen that UDA consistently outperforms GNL (Cao et al., 2023), particularly on Cartoon and Sketch domains. This suggests that, unlike DG methods, which aim to generalize to any unseen domain solely by training on the source domain, UDA can be more effective for semantic UAD since it exposes the model to the target domain during training, even if it is unlabeled.

**Feature extractor.** Table S9 compares several feature extractor backbones, showcasing the effectiveness of the proposed UDA method against source-only and the robustness of the methodology to the backbone used. Specifically, MobileNet-V2 (Sandler et al., 2018), as the smallest architecture, shows the weakest improvement, while ResNet18 (He et al., 2016) and ResNet50 (He et al., 2016) show better performance. Transformer-based models like CLIP-ViT-B32 (Radford et al., 2021) show an even higher performance. These results highlight the superiority of transformer models over CNN-based architectures for domain adaptation tasks on challenging datasets like VisDA (Peng et al., 2017), which comes at the cost of having a larger architecture.

Table S9. AUC performance on VisDA (Peng et al., 2017) of different trainable feature extractors using our method, in comparison against the source-only-trained model.

| | Our framework backbone composition: $\psi$: CLIP-ViT-B32 (fixed) | | | |
|---|---|---|---|---|
| Adapation setting | $\phi$: MobileNet-V2 | $\phi$: ResNet18 | $\phi$: Resnet50 | $\phi$: CLIP-ViT-B32 |
| Source only (DSVDD) | 55.63±06.06 | 56.80±10.90 | 62.83±8.60 | 81.89±17.38 |
| **Ours** (UDA) | **70.36**±10.00 | **72.47**±11.04 | **75.54**±11.57 | **85.92**±13.12 |

**Pretrained visual encoders.** As we are considering an artificial anomaly detection setting, i.e., in the training datasets, only one object class is considered as "normal" while all other classes are treated as "anomalies", our approach requires visual encoders that can effectively capture global object-level features. Hence, in Figure S9b, we compare our unsupervised model using various visual encoders against our source-only model, few-shot baselines (BiOST, MsRA), and an oracle (supervised) model. Among the visual encoders, CLIP (Radford et al., 2021) and CLIPSeg (Lüddecke & Ecker, 2022) exhibit the highest consistency and overall performance, with medians ranging between 75% and 80% AUC. SigLIP (Zhai et al., 2023) achieves comparable performance, though with slightly more variability. In contrast, Dino-v2 (Oquab et al., 2023) shows noticeably lower performance, suggesting that its representations may be less effective at capturing global object-level features. As expected,

the source-only model performs the worst, while the Oracle model reaches the highest and most stable performance. Compared to the baselines, our model with different visual encoders significantly outperforms the few-shot baselines, which have much lower performance.

Table S10. VisDA performance with different visual encoders. **w/o CLIP** means the $\phi$ is self-trained.

| AUC | Src Only | Few-shot | | Unsupervised (**Ours**) $\phi$ : R50 (fixed) | | | | |
|---|---|---|---|---|---|---|---|---|
| | | BiOST | MsRA | **w/o CLIP** | w/ CLIP | w/ SigLIP | w/ DINO-v2 | w/ CLIPSeg |
| Avg | 62.84 | 50.48 | 67.28 | 68.47 | 75.54 | 72.64 | 65.71 | **76.85** |
| ±std | ±8.55 | ±7.55 | ±5.79 | ±8.30 | ±9.80 | ±10.05 | ±13.71 | ±8.92 |

Table S11. Anomaly detection performance on the VisDA (Peng et al., 2017) dataset for the setting $f$:R50 + $\psi$:CLIP-ViT-B32 using additional metrics, such as Accuracy (Acc.), Balanced Accuracy (B.acc.), Precision (P), and Recall (R).

| Normal class | Source only (DSVDD) | | | | Few-shot (MsRA) | | | | Few-shot (**Ours**) | | | | Unsup. (**Ours**) | | | |
|---|---|---|---|---|---|---|---|---|---|---|---|---|---|---|---|---|
| | Acc. | B.acc. | P | R | Acc. | B.acc. | P | R | Acc. | B.acc. | P | R | Acc. | B.acc. | P | R |
| **CAD → Real** | | | | | | | | | | | | | | | | |
| Aeroplane | 65.54 | 63.62 | 95.66 | 65.86 | 62.17 | 66.09 | 96.44 | 61.51 | 77.28 | 74.46 | 97.21 | 77.75 | 83.42 | 77.35 | 97.35 | 84.44 |
| Bicycle | 56.40 | 61.36 | 96.42 | 55.74 | 45.20 | 55.61 | 95.54 | 43.81 | 66.33 | 67.72 | 97.17 | 66.15 | 72.49 | 73.61 | 97.87 | 72.34 |
| Bus | 53.91 | 62.83 | 94.90 | 51.78 | 56.11 | 59.71 | 93.54 | 55.26 | 59.82 | 67.26 | 95.86 | 58.04 | 75.59 | 75.30 | 96.59 | 75.66 |
| Car | 69.95 | 71.61 | 95.94 | 69.52 | 33.57 | 51.14 | 90.65 | 29.14 | 79.50 | 75.35 | 96.02 | 80.55 | 73.59 | 60.21 | 92.41 | 76.96 |
| Horse | 62.21 | 63.07 | 94.74 | 62.03 | 34.32 | 53.02 | 92.87 | 30.36 | 73.62 | 73.14 | 96.55 | 73.73 | 74.17 | 75.30 | 97.06 | 73.93 |
| Knife | 54.61 | 59.84 | 95.06 | 53.68 | 53.42 | 64.22 | 96.46 | 51.49 | 70.06 | 65.96 | 95.69 | 70.79 | 56.87 | 63.97 | 96.08 | 55.60 |
| Motorcycle | 63.33 | 64.39 | 93.61 | 63.02 | 47.28 | 56.13 | 91.64 | 44.73 | 69.50 | 72.70 | 95.92 | 68.58 | 79.56 | 77.61 | 96.24 | 80.13 |
| Person | 13.57 | 50.33 | 91.14 | 03.68 | 53.94 | 57.62 | 92.21 | 52.94 | 36.46 | 51.49 | 90.28 | 32.42 | 61.90 | 64.56 | 94.15 | 61.19 |
| Plant | 55.61 | 56.08 | 95.31 | 55.54 | 66.65 | 52.51 | 94.49 | 68.55 | 69.84 | 66.08 | 96.70 | 70.34 | 58.65 | 65.43 | 97.15 | 57.73 |
| Skateboard | 60.32 | 59.01 | 97.29 | 60.43 | 27.72 | 54.41 | 97.47 | 25.52 | 64.21 | 59.97 | 97.33 | 64.56 | 73.23 | 76.46 | 98.92 | 72.97 |
| Train | 47.36 | 53.90 | 91.53 | 45.71 | 48.84 | 53.48 | 91.30 | 47.67 | 59.23 | 57.60 | 92.33 | 59.64 | 67.33 | 64.70 | 94.04 | 67.99 |
| Truck | 59.72 | 58.87 | 93.23 | 59.92 | 71.18 | 52.56 | 91.23 | 75.46 | 66.94 | 67.28 | 95.25 | 66.86 | 74.84 | 58.05 | 91.82 | 79.07 |
| Avg. | 55.28 | 60.41 | 94.57 | 53.91 | 50.03 | 56.38 | 93.65 | 48.87 | 66.07 | 66.58 | 95.53 | 65.78 | **70.97** | **69.38** | **95.81** | **71.50** |
| ±std | 14.41 | 05.50 | 01.87 | 17.10 | 13.42 | 04.53 | 04.76 | 15.49 | 11.17 | 07.23 | 02.12 | 12.40 | 08.18 | 07.21 | 02.21 | 9.13 |

**Performance using additional metrics.** Table S11 presents the anomaly detection performance on the VisDA (Peng et al., 2017) dataset using additional metrics, including Accuracy (Acc.), Balanced Accuracy (B.acc.), Precision (P.), and Recall (R.). The results shown for our few-shot and unsupervised variants align with those presented in Table 2, particularly in terms of B.acc. and R.. Our few-shot variant outperforms MsRA (Li et al., 2023), demonstrating superior anomaly detection performance. This improvement may be attributed to using a contrastive alignment strategy, which explicitly maximizes the similarity between normal data in both source and target domains, as opposed to the implicit adversarial-based alignment used in MsRA (Li et al., 2023). The complexity of the VisDA (Peng et al., 2017) dataset, with its diverse backgrounds and resolutions in the target domain (Real), especially in comparison with other benchmarks, may also be an impacting factor in this performance drop. In contrast, our unsupervised approach benefits from the highest performance, underscoring the benefit of exposing the model to larger amounts of unlabeled data, which may result in better adaptation in complex datasets.

**Sensitivity analysis.** In Figure S9a, we investigate the impact of the two hyper-parameters $\lambda_1$ and $\lambda_2$ controlling the one-class optimization and domain alignment, respectively. Specifically, we set $\lambda_1$ to 1 and vary $\lambda_2$ to assess the impact of the adaptation loss. We find that the best performance is reached when both $\lambda_1$ and $\lambda_2$ are set to 1, indicating that the domain alignment objective is as important as the one-class classification during model optimization.

## B.2 QUALITATIVE RESULTS

**Histograms of anomaly scores.** The anomaly score distributions of the four methods (Source-Only, MsRA, Ours-Few-shot, and Ours) tested on all VisDA (Peng et al., 2017) classes are given in Figure S10. Overall, our few-shot and unsupervised methods better discriminate normal and anomalous target-domain samples compared to the Source-Only model. The few-shot variant shows a flatter anomaly distribution, likely due to the use of jigsaw-generated pseudo-anomalies, which closely resemble the original target normals. This may have led the model to focus on local changes in actual anomalies, resulting in a broader range of anomaly scores and less emphasis on global features.

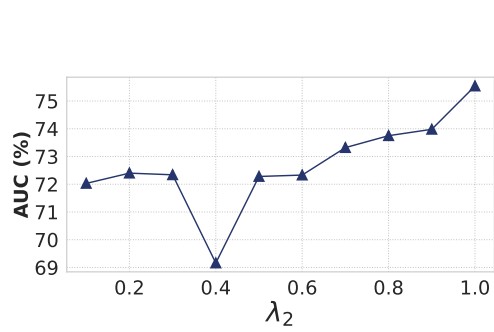

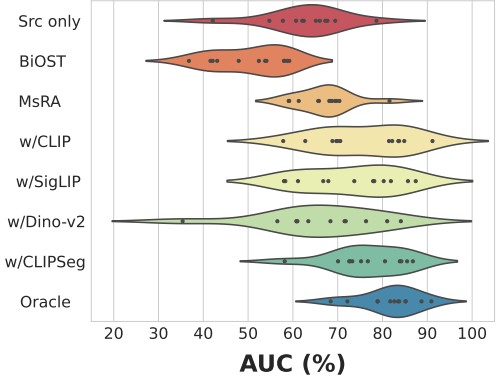

(a) Sensitivity analysis on VisDA when $\lambda_1 = 1$ (i.e., AD objective) and $\lambda_2$ varies (i.e., the UDA objective).

(b) Distribution of the AUC performance across the classes of VisDA of our unsupervised approach when using $\phi$ as ResNet50 coupled with various visual encoders $\psi$ vs. BiOST (Cohen & Wolf, 2019), MsRA (Li et al., 2023), and the Oracle.

Figure S9. ((a)) Sensitivity analysis on the VisDA dataset. ((b)) AUC comparison across different visual encoders.

In contrast, our unsupervised method exhibits a more peaked anomaly distribution. However, the incomplete separation of normal and anomalous scores suggests clustering limitations and highlights the need for filtering or noise removal mechanisms to better identify normal target samples.

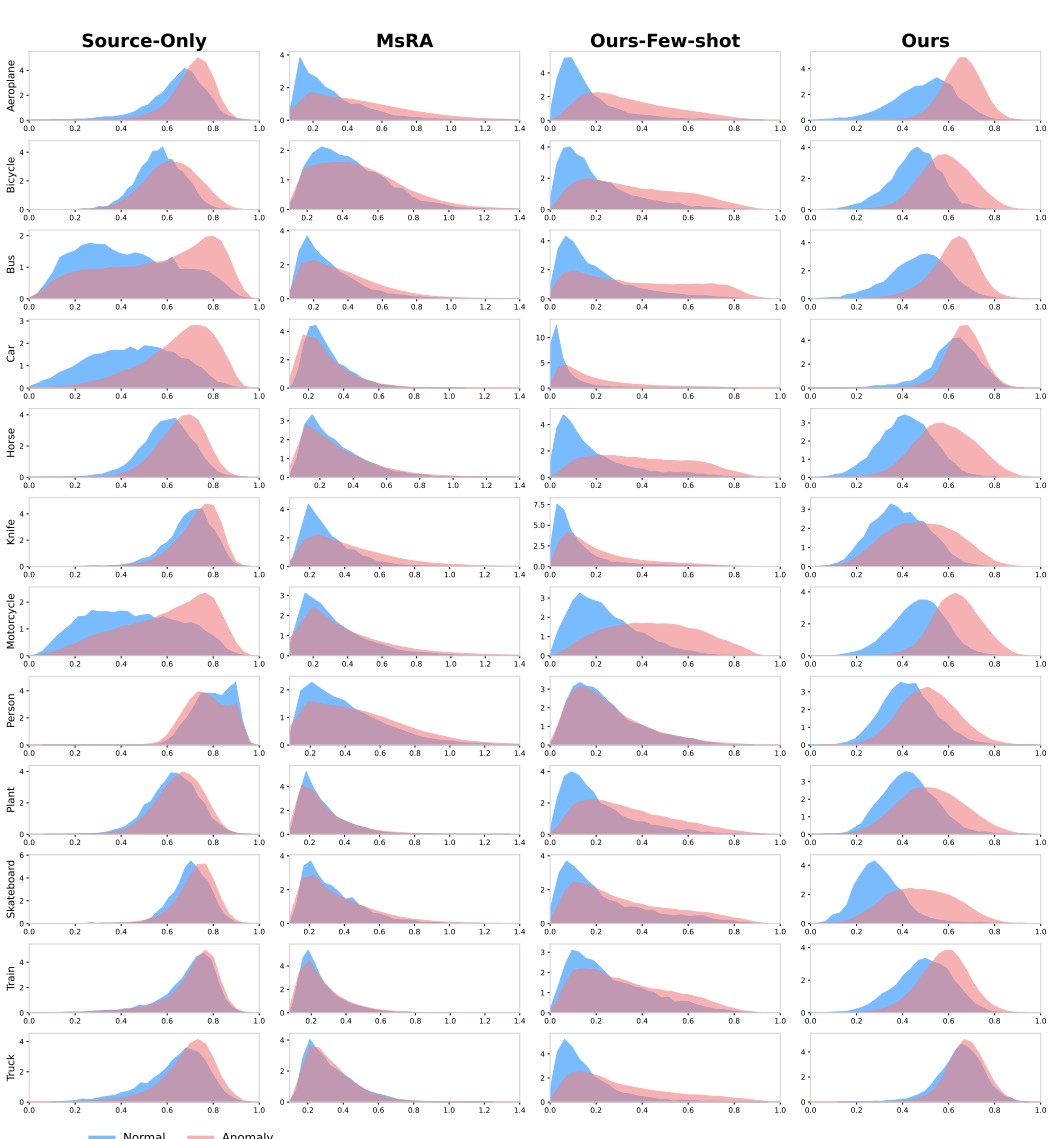

Figure S10. Histogram of anomaly scores for all classes of VisDA (Peng et al., 2017) (x-axis: anomaly score, and y-axis: count).

