# OpenReview forum: "When Unsupervised Domain Adaptation meets One-class Anomaly Detection: Addressing the Two-fold Unsupervised Curse by Leveraging Anomaly Scarcity"
_ICLR.cc/2026/Conference — Submitted to ICLR 2026_

### Official Review · Reviewer_iSVA · 2025-10-28

**Soundness:** 3
**Presentation:** 3
**Contribution:** 3
**Rating:** 6
**Confidence:** 3

**Summary:**

This paper introduces the first fully unsupervised domain adaptation (UDA) framework for one-class anomaly detection (UAD). Traditional UDA assumes access to class labels in the source domain and unlabeled samples in the target, but this setting becomes ill-posed in one-class anomaly detection where both tasks (UDA and AD) are unsupervised. The authors call this the “two-fold unsupervised curse.”

To address this, the paper proposes leveraging the scarcity of anomalies in the target domain. It assumes that normal samples form a dominant cluster in feature space, which can be identified using clustering on frozen CLIP visual features. The dominant target cluster is then aligned with source normal samples via a contrastive alignment strategy while optimizing a DSVDD (Deep SVDD) objective on the source domain.

Experiments on four UDA benchmarks (OfficeHome, Office31, VisDA, PACS) show consistent improvements over state-of-the-art few-shot and zero-shot baselines, with ablations confirming the importance of clustering and alignment components.

**Strengths:**

- The paper identifies and formalizes the "two-fold unsupervised curse" — the simultaneous unsupervised nature of both UDA and one-class AD. This conceptual contribution is well-motivated and fills a gap in the literature.

- The paper provides extensive experiments across multiple benchmarks, comparing against both few-shot and zero-shot adaptation methods. The results consistently show substantial performance gains in AUC

- The paper provides a clean problem setup, well-structured presentation, and promises to release code. Figures effectively illustrate the conceptual challenges and the intuition behind the solution.

**Weaknesses:**

- The method heavily relies on frozen CLIP features for dominant cluster discovery. This could be computationally expensive and less effective for non-visual or domain-specific data where CLIP representations are suboptimal.

- The core idea assumes that anomalies are rare enough for clustering to separate normal data. While realistic, this assumption limits applicability in scenarios with higher anomaly ratios or when anomalies are semantically close to normal samples.

**Questions:**

N/A

---

### Official Review · Reviewer_MK1J · 2025-10-31

**Soundness:** 3
**Presentation:** 3
**Contribution:** 3
**Rating:** 6
**Confidence:** 3

**Summary:**

This paper introduces the first framework for unsupervised domain adaptation (UDA) in unsupervised anomaly detection (UAD), addressing what the authors call the “two-fold unsupervised curse.” The key idea is to exploit the scarcity of anomalies by clustering the target domain’s features—extracted using a pretrained vision–language model (CLIP)—to identify a dominant (mostly normal) cluster and align it with source-domain normal features via contrastive learning. Experiments on standard UDA benchmarks demonstrate performance gains over source-only and few-shot baselines.

**Strengths:**

- The paper formulates a novel and well-motivated problem — UDA for one-class anomaly detection — that has not been explicitly studied before.
- The proposed solution (dominant cluster alignment via CLIP features and contrastive loss) is conceptually simple, modular, and effectively explained.
- The work bridges two important unsupervised paradigms (UAD and UDA) and sets a strong baseline for future research in cross-domain anomaly detection.
- The writing and structure are clear, with well-illustrated figures and appropriate contextualization of related work.

**Weaknesses:**

- The method heavily relies on the assumption that the dominant target cluster corresponds to normal data. It is difficult to transfer to multi-classification tasks or other type of UAD tasks.

- The approach depends on cluster count (K) and CLIP feature choices, with no unsupervised guidance for tuning.

**Questions:**

- How sensitive is the method to the choice of the pretrained model? Would self-supervised visual encoders (e.g., DINOv2) yield similar clustering behavior?

- Given CLIP’s role, could large multimodal language models (e.g., LLaVA, GPT-4V) help verify or refine cluster purity through semantic reasoning?

---

### Official Review · Reviewer_Peru · 2025-11-01

**Soundness:** 2
**Presentation:** 3
**Contribution:** 2
**Rating:** 4
**Confidence:** 4

**Summary:**

This paper introduces a framework for unsupervised domain adaptation (UDA) for one-class unsupervised anomaly detection (UAD). The proposed method uses a dual-branch architecture: (1) a frozen, pre-trained visual encoder to cluster the target data and identify the dominant (pseudo-normal) cluster, and (2) a separate, trainable feature extractor. This trainable model is optimized with a dual objective: a one-class DSVDD loss on the source data and a contrastive UDA loss. This UDA loss works to align the source features with the features of the dominant target cluster while simultaneously pushing away features from non-dominant clusters. The authors evaluate their method on several standard UDA benchmarks (Office-Home, VisDA, PACS) adapted to a one-vs-all semantic anomaly detection protocol. Their results demonstrate state-of-the-art performance.

**Strengths:**

1. The paper addresses the unsupervised domain adaptation for one-class unsupervised anomaly detection, which is a challenging and practical scenario.
2. The paper is well written and easy to follow.
3. The paper provides extensive experiments, showing the effectiveness and versatility of the proposed method.

**Weaknesses:**

1. The proposed loss (eq 9) is a simple combination of the loss in DSVDD and standard contrastive loss. The novelty and insight are limited.

2. The proposed method uses a strong extra CLIP model that can already extract good features to separate anomalies, which is unfair to compare to the baselines (i.e., compared with BiOST, TSA, ILDR, etc. in Table 1, where they don't use a strong model). As shown in Table 2, the original CLIP model without adaptation is better than the Few-shot adaptation version of the proposed method.

3. The method relies on a hard-assignment K-means clustering to label target samples as either dominant (pseudo-normal) or non-dominant (pseudo-anomalous). This hard assignment seems brittle. 1) If the target normal data is inherently multi-modal (e.g., two different types of normal 'bikes'), K-means might identify only one as "dominant" and incorrectly label the other normal mode as "non-dominant." The model would then be explicitly trained to push this valid normal data away, actively harming performance. 2) If the dominant cluster contains some anomalies (or vice-versa), this hard assignment will introduce label noise that is directly propagated into the contrastive loss.

4. The number of clusters K is a key hyperparameter that is set manually per dataset (K=2, 5, or 10).

**Questions:**

1. Could the authors clarify the exact, unsupervised methodology used to select the number of clusters (K) for each dataset? If it was tuned, how can this be justified in a fully unsupervised setting? How badly does performance degrade if a wrong K is chosen (e.g., K=2 for VisDA, or K=10 for Office)?

---

### Official Review · Reviewer_YDzx · 2025-11-01

**Soundness:** 1
**Presentation:** 3
**Contribution:** 1
**Rating:** 2
**Confidence:** 4

**Summary:**

The paper presents a framework for unsupervised domain adaptation (UDA) in unsupervised anomaly detection (UAD). It addresses performance degradation under domain shift and formalizes the challenge as the two-fold unsupervised curse, wherein naïve adaptation may align normal source data with both normal and anomalous samples in the unlabeled target domain. To mitigate this, the method assumes anomalies are scarce and clusters target features to identify a dominant, likely-normal cluster. It then aligns only that cluster with the one-class normal source features. Target features for clustering are extracted with a frozen CLIP visual encoder, while a trainable feature extractor is learned on source normals using Deep SVDD (DSVDD); a contrastive loss encourages cross-domain alignment.

**Strengths:**

Designing robust anomaly detectors that generalize well to new domains is important, and the paper states its goals and contributions clearly.

**Weaknesses:**

[A] Robust Novelty Detection through Style-Conscious Feature Ranking

[B] A Contrastive Teacher-Student Framework for Novelty Detection under Style Shifts

[C] Deep Semi-Supervised Anomaly Detection

[D] Deep Nearest Neighbor Anomaly Detection


[E] PANDA: Adapting Pretrained Features for Anomaly Detection and Segmentation


[F] CSI: Novelty Detection via Contrastive Learning on Distributionally Shifted Instances


W1) The authors motivate the problem with large real-world datasets (e.g., medical), but the paper does not include any challenging open-world datasets to support this claim.

W2) Industrial use cases often involve texture/pixel-space anomalies; well-known benchmarks such as MVTec AD are absent from the evaluation.

W3) There exist closely related works (e.g., [A, B]) that are neither cited nor discussed. The authors should reference these methods and compare against them within the proposed pipeline.

W4) The Deep SVDD objective is known to be vulnerable to collapse (e.g., mapping features to a constant) [C]. How do the authors prevent this failure mode in practice?


W5)  Leveraging pretrained features in combination with an SVDD-style loss is not novel [E], and both k-means [D] and contrastive objectives [F] are standard baselines. The paper should clarify its technical novelty and delineate contributions beyond prior work.

W6) Code is not available, which hinders reproducibility.

**Questions:**

See Weaknesses.

---

### Meta-Review · Area_Chair_z3Cc · 2025-12-30

**Summary:**

The major concerns of the reviewers are the limited novelty, too strong assumption, and unfair experimental comparison. The authors haven't provided any rebuttal.  After carefully reading the paper, I agree with the major concerns raised by the reviewers and believe that the paper falls short of the bar for acceptance. Therefore, I recommend rejection.

**Reviewer Concerns:**

**Reviewer YDzx**'s major concerns
1) Several related works and baselines haven't been discussed and compared with the proposed method in the experiments.
2) The novelty of the proposed method is not significant compared with prior works.

**Reviewer Peru**'s major concerns
1) The novelty and insight are limited.
2) The comparison with BiOST, TSA, and ILDR is not fair due to the use of CLIP in the proposed framework.
3) The effectiveness and rationality of using k-means to find the dominant cluster containing normal data haven't been sufficiently justified.

**Reviewer MK1J**'s major concerns
1) The assumption of a dominant target cluster is too restrictive.

**Reviewer iSVA**'s major concerns
1) The method heavily relies on CLIP features and will be less effective for non-visual or domain-specific data where CLIP representations are suboptimal.

Since the authors haven't provided any responses to the reviewers' comments, I assume that all of these concerns remain.

**Reviewer Scores:**

**Reviewer YDzx** and **Reviewer Peru** may keep their negative assessments.

---

### Decision · Program_Chairs · 2026-01-26

Reject